# Chickpea Roots Undergoing Colonisation by *Phytophthora medicaginis* Exhibit Opposing Jasmonic Acid and Salicylic Acid Accumulation and Signalling Profiles to Leaf Hemibiotrophic Models

**DOI:** 10.3390/microorganisms10020343

**Published:** 2022-02-02

**Authors:** Donovin W. Coles, Sean L. Bithell, Meena Mikhael, William S. Cuddy, Jonathan M. Plett

**Affiliations:** 1Hawkesbury Institute for the Environment, Western Sydney University, Richmond, NSW 2753, Australia; d.coles@westernsydney.edu.au; 2NSW Department of Primary Industries, Tamworth, NSW 2340, Australia; sean.bithell@dpi.nsw.gov.au; 3Mass Spectrometry Unit, School of Medicine, Western Sydney University, Campbelltown, NSW 2560, Australia; m.mikhael@westernsydney.edu.au; 4NSW Department of Primary Industries, Elizabeth Macarthur Agricultural Institute, Menangle, NSW 2568, Australia; will.cuddy@dpi.nsw.gov.au

**Keywords:** *Cicer arietinum*, disease development, pathogenesis, phytohormone, plant immunity, transcriptomics

## Abstract

Hemibiotrophic pathogens cause significant losses within agriculture, threatening the sustainability of food systems globally. These microbes colonise plant tissues in three phases: a biotrophic phase followed by a biotrophic-to-necrotrophic switch phase and ending with necrotrophy. Each of these phases is characterized by both common and discrete host transcriptional responses. Plant hormones play an important role in these phases, with foliar models showing that salicylic acid accumulates during the biotrophic phase and jasmonic acid/ethylene responses occur during the necrotrophic phase. The appropriateness of this model to plant roots has been challenged in recent years. The need to understand root responses to hemibiotrophic pathogens of agronomic importance necessitates further research. In this study, using the root hemibiotroph *Phytophthora medicaginis*, we define the duration of each phase of pathogenesis in *Cicer arietinum* (chickpea) roots. Using transcriptional profiling, we demonstrate that susceptible chickpea roots display some similarities in response to disease progression as previously documented in leaf plant–pathogen hemibiotrophic interactions. However, our transcriptomic results also show that chickpea roots do not conform to the phytohormone responses typically found in leaf colonisation by hemibiotrophs. We found that quantified levels of salicylic acid concentrations in root tissues decreased significantly during biotrophy while jasmonic acid concentrations were significantly induced. This study demonstrated that a wider spectrum of plant species should be investigated in the future to understand the physiological changes in plants during colonisation by soil-borne hemibiotrophic pathogens before we can better manage these economically important microbes.

## 1. Introduction

Plant pathogens display diverse infection strategies during colonisation of a plant host. Biotrophic pathogens such as *Puccinia striiformis* f.sp. *tritici*, the causal agent of wheat stripe rust, require a living host to complete their lifecycle [1,2]. Conversely, necrotrophic pathogens such as *Botryits cinerea*, the cause of grey mould on a broad range of host species, including chickpea, kill host tissues during the colonisation process to obtain nutrients from dead tissue [3]. Between these two opposing lifestyles are hemibiotrophic pathogens such as *Phytophthora infestans* which display an initial biotrophic colonisation phase followed by a physiologically regulated switch to a necrotrophic phase during later phases of infection [4,5]. During the biotrophic phase, hemibiotrophs can invade their hosts both intercellularly and intracellularly through the formation of haustoria such as in the soybean root pathogen *Phytophthora sojae* or they can remain strictly intercellular between mesophyll cells as observed for *Leptosphaeria maculans*, the phoma stem canker pathogen of canola [6,7]. During this biotrophic phase of infection, effector proteins are released to suppress plant defences while nutrients are obtained from living cells or the apoplastic space [8]. Subsequently, a major morphological and physiological switch to the necrotrophic phase occurs (henceforth termed the ‘biotrophy to necrotrophy switch’; BNS), whereby hemibiotrophs begin to kill plant cells through the release of toxins and necrosis-inducing proteins, and digest on dead cells [9,10].

The plant immune response to hemibiotrophic pathogens before and after the switch from biotrophy to necrotrophy is mediated by a series of intracellular signalling cascades that are regulated, in part, by plant phytohormones [11,12]. The phytohormones salicylic acid (SA), jasmonic acid (JA), and ethylene (ET) all play a role during plant–microbe interactions, the relative importance of each depending upon the lifestyle of the pathogen colonising the plant tissue [13]. SA is known as a positive regulator of defence against biotrophs, whereas JA/ET act synergistically as positive regulators of defence against necrotrophy [12,14]. Induction of the SA pathway leads to transcription of *NONEXPRESSOR OF PATHOGENESIS-RELATED GENES1* (*NPR1*), which in turn leads to the production of antimicrobial PATHOGENESIS-RELATED (PR) proteins PR1, PR2 and PR5 locally and systemically [15,16]. SA plays a role during both pattern-triggered immunity and effector-triggered immunity, and in systemic defence response called systemic acquired resistance [17]. Conversely, the JA pathway activates the induction of plant defensin 1.2 (PDF1.2), PR3, PR4 and PR12, which accumulate locally [12,15]. JA and ET function in both resistance to necrotrophic pathogens and wound responses [18,19]. Accordingly, when a hemibiotrophic pathogen is encountered, a biphasic hormonal response ensues, with SA observed during the biotrophic phase and JA/ET during the necrotrophic phase [20,21,22]. 

The above-mentioned phytohormone response model stems mainly from research aimed at deciphering the molecular response of foliar tissues [11,23,24,25,26,27,28]. From these model systems, it is evident that defence induction involving pathogen recognition receptors and receptor-like kinases, and alterations in primary metabolism is among some of the responses specific to the biotrophic phase. Conversely, cell wall lignification, secondary metabolism and cell death-associated pathways are some of the responses specific to the necrotrophic phase. The molecular mechanisms underlying plant defences in leaves and stems are better understood than for below-ground root responses to soil-borne hemibiotrophic pathogens. Previous work has postulated that root responses to pathogens differ to those induced in leaves [29,30,31]. However, some similarities in root responses to those of leaves have been observed during similar phases of hemibiotrophic infection. For instance, during biotrophic growth of *P. parasitica* in *Arabidopsis* roots, both primary metabolism, e.g., glycolysis and receptor kinase were observed to be down-regulated [32]. Primary metabolism is also altered in *Arabidopsis* during this phase of infection by *Verticillium dahlia* with amino acid metabolism observed down-regulated while sugar metabolism was up-regulated [33]. During the necrotrophic phase, induction of secondary metabolism related to phytoalexin and defence-related gene expression have been observed in the soybean/*P. sojae* pathosystem [34,35]. While some similarities in root responses to leaf responses exist, the role played by phytohormones in roots is less understood. During infection of *Arabidopsis* roots by *Phytophthora parasitica*, SA and JA responses are both activated transiently during penetration but down-regulated during further invasive growth, whereas ET responses were up-regulated during these phases [36]. In addition, *Arabidopsis* roots infected with root hemibiotroph *Fusarium oxysporum* display an induction of JA and ET pathways while SA pathways are repressed during both biotrophic and BNS phases [30]. This suggests that root responses to hemibiotrophic pathogens may not reflect results found in foliar pathosystems. 

With the increasing number of sequenced plant genomes, new model systems should be considered to broaden our understanding of root responses to hemibiotrophic pathogens, especially within agriculturally relevant systems. An emerging model for root hemibiotrophic pathogenesis is the interaction between *Cicer arietinum* (chickpea) and *Phytophthora medicaginis* [37,38]. Chickpea is an agronomically important crop worldwide due to its high nutritional value and in developing countries, where it provides an inexpensive source of protein [39]. *P. medicaginis* is a root rot pathogen of chickpea causing losses of >50% during above-average rainfall conditions [40]. During early phases of infection, *P. medicaginis* hyphae grow intercellularly while haustoria develop intracellularly, suggesting an initial phase of biotrophy as has been observed for other root *Phytophthora* spp. such as *P. sojae* [41,42]. Later in infection, *P. medicaginis* then switches to cause root rot associated with hyphal ramification and sporulation within diseased roots [43,44,45]. To date, the timing of the phases of *P. medicaginis* hemibiotrophic infection in chickpea roots have not been mapped out. The availability of the chickpea Desi genome has provided a valuable resource to decipher the plant response to abiotic stress, i.e., drought and salt stress [46] and, thus far, the defence response of chickpea to *P. medicaginis* has only been reported for early stages of infection [37,47]. The enriched pathways in susceptible chickpea var. ‘Sonali’ that were repressed at this stage of infection included secondary metabolism and the hormone biosynthetic process while glucose homeostasis and response to ethylene stimulus were examples of induced pathways [37]. In susceptible chickpea var. ‘Rupali’ during early infection by *P. medicaginis*, some of the enriched pathways that were induced included catalytic activity, receptor activity and cell communication which were absent in resistant [47]. 

Using the *C. arietinum*/*P. medicaginis* root pathosystem, we sought to define the timing of the lifestyle switches from biotrophy to BNS to necrotrophy and then investigate the plant response during each of these phases. We chose to use a microcosm-based system to improve our control of the inoculation and to improve the reproducibility of the timing of the three phases. Based on previous research considering plant responses to hemibiotrophic pathogens, we initially hypothesized that: 1) defence responses would be suppressed during the biotrophic phases and induced during the necrotrophic phase, and 2) chickpea would display a biphasic response involving SA and JA during the biotrophic and necrotrophic phases, whereby SA would be highest during biotrophy and JA would be highest in necrotrophic tissues. Our results only partially supported these hypotheses, suggesting that a wider spectrum of plant–hemibiotrophic interactions need to be considered to better understand the basis of plant susceptibility to hemibiotrophic pathogens.

## 2. Materials and Methods

### 2.1. Plant Material and Phytophthora Inoculations

Seeds of the *P. medicaginis* root rot-susceptible Desi chickpea var. ‘Sonali’ were obtained from the New South Wales Departments of Primary Industries (NSW DPI, Tamworth, NSW, Australia). Seeds were surface sterilised (4% NaOCl solution for 15 min, 70% EtOH for ten minutes and sterile ddH_2_O rinse three times) before sowing on modified full-strength Murashige and Skoog agar plates (4.4% MS; 3% CaCO_3_, pH 7, 1.1% bacto agar) supplemented with Gamborg’s vitamins. Seedlings were grown in controlled growth chambers with a 15 h light/9 h dark cycle at 18 °C, 70% relative humidity and 500 μmol m^−2^ s^−1^ light intensity. Fifteen-day-old seedlings were used for inoculation. 

*P. medicaginis* isolate TR7831 was isolated from soil collected from a chickpea paddock by the New South Wales Department of Primary Industries, Tamworth, Australia, and propagated on V8 juice agar plates (containing 200 mL L^−1^ V8^TM^ bottled juice; 3.0 g L^−1^ CaCO_3_, 15.0 g L^−1^ agar) supplemented with ampicillin (100 μg mL^−1^) at 25 °C in the dark. Prior to inoculations, *P. medicaginis* was passaged through the host to ensure pathogenicity, and after reisolating on V8 juice agar plates supplemented with ampicillin (100 ug mL^−1^), the second re-culturing of *P. medicaginis* was used to prepare inoculum. *P. medicaginis* inoculum (third re-culturing) was prepared one day before inoculations by aseptically transferring mycelial agar blocks taken from the edge of an actively growing culture onto 0.5 × 0.5 mm^2^ sterile membranes overlaid on V8 juice agar plates. For mock plant inoculations, agar blocks without mycelia were transferred onto 0.5 × 0.5 mm^2^ sterile membranes blocks overlaid onto V8 agar plates. For *P. medicaginis* axenic cultures, agar blocks carrying mycelium were transferred onto membranes overlaid on V8 agar plates. All plates were incubated for 12 h overnight at 25 °C in the dark. For inoculations, membranes carrying agar blocks with *P. medicaginis* mycelium were placed in direct physical contact with the root (to ensure a definitive “0 h” time for all experiments), at a point 1.5 cm from the tip of fifteen-day-old seedlings. For mock inoculations, membranes carrying agar blocks without mycelium were placed as above onto the roots of seedlings. After inoculations, seedlings were replaced into growth chamber using the same conditions as above.

### 2.2. Electrolyte Leakage Root Cell Death Assay

Relative electrolyte leakage from *P. medicaginis* inoculated and mock-inoculated chickpea roots was performed according to the method by [48] with modifications. Briefly, two 1.5 cm root segments, with the *P. medicaginis* inoculated site centred in the segment, were harvested alongside mock-inoculated roots at 12, 24, 36, 48, 60, 72 and 120 h post-inoculation (hpi) and submerged in 5 mL of ddH_2_0 for one h. The two 1.5 cm segments constituted a single sample unit/biological replicate. The electrolyte conductivity of three technical replicates per plant were measured using an Ohaus 300C portable conductivity meter with an IC-STCON3 electrode (Instrument Choice, Adelaide, Australia), with these samples taken from six control and six inoculated plants at each timepoint (i.e., 6 biological replicates per timepoint). Samples were then placed at 95 °C for one h to lyse all cells. After cooling samples to room temperature, electrolyte conductivity was again measured for all samples. The relative electrolyte conductivity was determined by calculating the percentage of electrolyte leakage before boiling relative to post-boiling of root samples. Larger ion conductivity in pre-boiled samples indicated greater cell disruption/death. The relative electrolyte leakage values were converted to a proportion and arcsine transformed due to the uneven variance of the non-transformed data. The data were analysed using ANOVA, with values for each plant considered a replicate for the factors time in hpi and *P. medicaginis* treatment (inoculated or mock-inoculated control). All analyses were conducted in GenStat [49]. Significant difference between inoculated and mock-inoculated control was determined using Least significant difference (LSD). The data were visualised in R v.4.0.0 [50].

### 2.3. Macroscopic Assessment of Symptoms

Macroscopic observation of symptoms in roots of chickpea seedlings following *P. medicaginis* inoculation and mock inoculation was performed every 12 h over five days as above. The biotrophic phase was defined as the time from inoculation until just before the first appearance of water soaking. Water soaking is when a wet, translucent, and sunken appearance appears on the root at the site of inoculation. The period during which a dark brown necrotic lesion was observed was defined as the necrotrophic phase. The ‘switch’ phase was defined as the period between the first appearance of water soaking and the first appearance of necrotic lesions in combination with electrolyte leakage analysis as described above. None of the images taken at the mentioned timepoints were modified.

### 2.4. Root Cell Viability Analysis

Chickpea var. ‘Sonali’ root segments of 1.5 cm in length were harvested from an area of the root surrounding the *P. medicaginis* inoculated site at 12, 24 and 72 hpi, alongside mock-inoculated site. All roots were washed three times with 1× phosphate-buffered saline (PBS; pH 7.4) and stained with a combined fluorescein diacetate (FDA; 10 µg mL^−1^) and propidium iodide (PI; 50 µg mL^−1^) live/dead cell staining solution for 15 min. The roots were washed three times with 1x PBS (pH 7.4) for 3 min at each wash and observed by confocal microscopy using a TCS SP5 confocal laser scanning microscope (Leica, Sydney, Australia). To detect FDA, the argon laser was used with an excitation wavelength of 488 nm and emission detection wavelengths of 500–550 nm. Detection of PI was carried out with a 561 DPSS laser with an excitation wavelength of 561 nm and emission detection wavelengths of 580–660 nm. Using a 20× objective the number of living cells and dead cells were counted in one field of view, the first layer of epidermal cells. The proportion of living to dead cells were determined for each timepoint and mock-inoculation. Data visualisation was performed in excel and statistical analysis of data was performed using ANOVA with the Tukey method and car package in R v.4.1.0 [51,52]. Post hoc comparison among groups was performed using the emmeans package in R v4.1.0 [51,53].

### 2.5. Propidium Iodide/Wheat Germ Agglutinin Staining and Confocal Microscopy

Root segments of 1.5 cm in length surrounding the inoculated site were harvested from *P. medicaginis* inoculated and mock-inoculated seedlings at 12, 24, and 72 hpi as above and fixed in 4% paraformaldehyde (Sigma-Aldrich, Castle Hill, Australia) at 4 °C. A portion of roots from each treatment were further rinsed three times with 1× PBS (pH 7.4) and cut into approximately 0.5 cm portions before embedding in 6% agarose and stored at 4 °C for 24 h, after which they were sectioned. Longitudinal root sections (30 µm in thickness) were cut using a vibratome 7000smz-2 (Campden Instruments, Lafayette, Indiana) and stored in 1× PBS (pH 7.4) at 4 °C until staining. For staining, roots were rinsed three times with 1× PBS (pH 7.4) prior to co-staining with Wheat Germ Agglutinin labelled fluorescein isothiocyanate (WGA-FITC; 100 µg mL^−1^) for 15 min and PI (20 µg mL^−1^) for 10 min. WGA-FITC stains the cell walls of certain oomycetes while PI stains the cell walls of plant cells [54,55]. The roots were washed 3 times with 1× PBS (pH 7.4), then transferred to new PBS (pH 7.4) and stored at 4 °C in the dark until analysis by confocal microscopy. Confocal microscopy was carried out on a TCS SP5 confocal laser scanning microscope (Leica, Sydney, Australia). To detect WGA-FITC, the argon laser was used with an excitation wavelength of 488 nm and emission detection wavelengths of 500–550 nm. Detection of PI was carried out with a 561 DPSS laser with an excitation wavelength of 561 nm and emission detection wavelengths of 580–660 nm. Z-stack images were taken to generate a 3D reconstruction of the inoculated site with a Z step size of 0.5–1 and frame average of 3. 

### 2.6. RNA-Sequencing and Data Analysis

Root segments of 1.5 cm in total length from above and below the inoculated site were harvested from *P. medicaginis* inoculated seedlings at 12, 24, and 72 hpi, and mock-inoculated controls before snap freezing samples in liquid N_2_ and storing at −80 °C. *P. medicaginis* axenic cultures that had been growing on modified full-strength MS media (same media as used for chickpea growth and pathogenesis experiments) for 12 h were harvested, snap frozen in liquid N_2_ and stored at −80 °C. RNA was extracted from four independent biological replicate samples for inoculated and mock-inoculated controls using the Bioline Plant II RNA extraction kit (Bioline, Eveleigh, Australia) according to the manufacturer’s guidelines. Poly-A RNA libraries were prepared and sequenced by GENEWIZ (Suzhou, China) using an Illumina HiSeq 2000 platform and 150 bp paired-end configuration. Raw RNA-seq reads were trimmed to remove low-quality sequences and adapters, and then aligned to the primary transcripts of the *Cicer arietinum* Desi genome [46] using CLC Genomics Workbench v.10.0.0 (Qiagen, Victoria, Australia). For the alignment, the minimum length fraction was set to 0.9, the minimum similarity fraction 0.9 and the maximum number of hits per read was 10. The total read counts for each transcript was determined and normalised to reads per kilobase of transcript, per million mapped reads using the RNA-Seq Analysis function in CLC Genomics Workbench v.10.0.0 (Qiagen, Victoria, Australia). Only transcripts that had an average normalised count of more than 10 in at least one of the timepoints or control were considered for further analysis. The Bioconductor package DESeq2 v.1.26.0 [56] was used to normalise raw transcript counts and identify statistically significant *C. arietinum* differentially expressed genes (DEGs) using the Benjamini–Hochberg test for multiple testing with a false discovery rate (FDR) to control for false positives and negatives. A principal component analysis (PCA) was performed using the *C. arietinum* transcripts to assess the variability between samples in R v.4.0.0 [50]. Only transcripts with a log2-fold change of −1 < log2FC > 1 compared to control and FDR-corrected *p* < 0.05 were kept for further analysis. The number of significantly differentially regulated genes at each of the timepoints was determined and visualised using R v.4.0.0 [50]. The log2-transformed data of differentially regulated genes as compared to control that were significant in at least one of the timepoints were visualised by heatmap using Morpheus and hierarchical clustered using Euclidean distance by genes according to the expression patterns (https://software.broadinstitute.org/morpheus/, accessed on 25 April 2020). 

### 2.7. Quantitative PCR Expression Validation of RNA Sequencing 

Root samples from the timecourse of colonisation were snap frozen in liquid nitrogen and their RNA extracted as described above. RNA transcript quantitation by Q-PCR was used to verify that a selection of genes showing different expression patterns were differentially regulated as expected. Briefly, total RNA (1 μg) was used for generation of cDNA using the Tetro cDNA Synthesis kit using only the Oligo dT_18_ primer (Bioline, Sydney, Australia). Q-PCR performed using 5× diluted cDNA and Bioline SensiFast no ROX Q-PCR mix using the primers detailed in Appendix A. Two genes were utilized to normalise all results (*Ca_00593*, *Ca_08899*). At the end of each Q-PCR run, a dissociation curve assay (from 95 to 65 °C) was performed to ensure the specificity of each reaction. A total of three biological replicates per treatment were used to determine expression levels of each gene. Relative expression was determined using the 2^−ΔΔCT^ method, whereby mock-inoculated chickpea roots were used as a control for the timecourse.

### 2.8. Gene Ontology (GO) Enrichment Analysis of the Unique Gene Sets at Each Phase 

Significantly DEGs were identified from the above analysis for the timepoints 12, 24, and 72 h post-*P. medicaginis* inoculation to identify unique responses during successive phases in disease progression. Gene lists that were up-regulated or down-regulated in each of the timepoints were uploaded into Venny v.2.1.0 (https://bioinfogp.cnb.csic.es/tools/venny/, accessed on 25 April 2020) to identify the genes that were uniquely regulated in each of the timepoints. The uniquely regulated gene lists for each of the timepoints were then combined and used to perform GO enrichment using the TopGO v.2.36.0 package in R v.4.0.0 [50,57]. Enrichment was also performed separately for the up-regulated and down-regulated gene lists at each of the timepoints. The classic algorithm was used to run the GO enrichment test in the biological process category. Enrichment was performed with the *C. arietinum* Desi genome as the gene universe, and the gene-to-GO map file used was generated from the *C. arietinum* annotation file in R. Fisher’s exact test was used to calculate *p*-value’s and the topNodes parameter was set to 50 to return the top 50 GO terms. Only GO terms with a *p* < 0.05 were used for further investigation. The top 20 GO terms for each timepoint with the number of significant genes and *p*-values for each term were visualised using R v.4.0.0 [50].

### 2.9. Expression Analysis of Chickpea Hormone Pathway Related Genes

Genes relating to the biosynthesis, signalling and downstream response for defence phytohormones, SA, JA and ET were identified from previous publications and the chickpea Desi genome annotation [46]. We also validated a selection of these annotations within the hormone perception/signalling pathways through BLAST search of the chickpea genome with *Arabidopsis* genes (Appendix A). Only those genes that were differentially regulated as compared to the control were used for further analysis. The log2-transformed data of the differentially regulated phytohormone-related genes as compared to control were visualised by heatmap using Morpheus (https://software.broadinstitute.org/morpheus/). 

### 2.10. In Vivo Root Salicylic and Jasmonic Acid Measurements

Roots from the same timepoints used for RNA sequencing were snap frozen in liquid N_2_ and stored at −80 °C prior to hormone extraction. SA and JA were extracted from four replicate samples for both inoculated and control seedlings according to the method of Hall et al. 2019 with modifications. Briefly, approximately 50 mg ground root tissue per sample was extracted with 70% methanol spiked with [^2^H_4_]-Salicylic acid (Olchemim, Olomouc, Czech Republic), Jasmonic-d_5_ Acid (2,4,-d_3_; acetyl-2,2-d_2_; CND isotopes, Quebec, Canada), and [^2^H_2_]N-[(-)-Jasmonoyl-Isoleucine (OIChemIm, Czech Republic), to yield a final concentration of 100 ppb. Subsequently, the samples were mixed using a rotor mixer at 80 rpm for 30 min at 4 °C and then centrifuging at 12,000 rpm for 5 min at room temperature. The extracts were analysed by UPLC/ESI-MS/MS using Acquity UHPLC coupled to a Xevo triple quadrupole mass spectrometer (Waters Corporation, Milford, Massachusetts). Five microlitres of extract were injected into a 2.1 mm × 50 mm × 1.7 µm, C18 reverse phase column. To identify SA, JA, and JA-Ile, comparisons were made to the fragmentation pattern of pure standards. A standard calibration curve was used to quantify the amount of each hormone and adjusted for sample recovery as compared to the internal standard. Fresh weight of each sample was used to standardise the final concentrations, and data were analysed and visualised using R v.4.0.0 [50].

## 3. Results

### 3.1. Phytophthora Medicaginis Infection of Chickpea Roots Features an Early Switch from Biotrophy to Necrotrophy 

To establish the timeline during which *P. medicaginis* exhibited a biotrophic phase, and when the switch occurred to necrotrophy in the Desi chickpea var. ‘Sonali’, we used four methods: macroscopic assessment of symptoms on roots, microscopic evaluation of *P. medicaginis* development, ion leakage analysis, and cell viability stains to determine when significant root cell death occurred. From the macroscopic assessment of chickpea roots, no symptoms were observed at the inoculated site until after 12 hpi as compared to the mock-inoculated control (Figure 1a,b). At 24 hpi, water soaking was evident, whereby roots exhibited a wet, sunken, and translucent appearance in the infected area (Figure 1c). By 72 hpi, a dark brown lesion was observed which indicated that necrosis had occurred at the inoculated site (Figure 1d). Invading hyphae of *P. medicaginis* stained green with fluorescently tagged WGA were observed penetrating the cortical cell layer of chickpea roots at 12 hpi with haustoria-like penetrating root cells indicated by white arrows (Figure 1e,f). At 24 hpi, *P. medicaginis* hyphae were observed penetrating deeper into the cortical cell layer of chickpea roots with both intracellular and intercellular hyphal growth visibly present (Figure 1g). Furthermore, haustoria-like structures indicated by white arrows were also observed within root cells at 24 hpi (Figure 1g,i). By 72 hpi intercellular and intracellular hyphal ramification and sporulation (blue arrows) was observed between plant cells and root surface (Figure 1h,j). 

Plant cell death assays, based on the percentage of ion leakage from the roots, were performed to quantitatively verify when *P. medicaginis* switched to the necrotrophic phase. A baseline level of ~25% electrolyte leakage was observed in the control (Figure 2a), likely due to the excision sites from where the root was removed from the plant. We observed a similar level of leakage until 36 hpi after which there was significantly higher electrolyte leakage in the *P. medicaginis* inoculated roots compared to mock-inoculated tissues indicating the necrotrophic phase of infection had begun (Figure 2a). At 24 hpi, non-significant, but higher, electrolyte leakage was observed in *P. medicaginis* inoculated roots (Figure 2a), suggesting that this was the timepoint at which *P. medicaginis* began to switch from biotrophy to necrotrophy in our experimental system. We then performed cell viability staining of roots 12, 24, and 72 hpi to validate the timepoints for the phases of hemibiotrophic infection. The percentage of cell viability did not significantly differ between the control, 12 and 24 h timepoints post-inoculation indicating the cells were still living at these timepoints (*p* > 0.05, Figure 2b). While non-significant, cell viability was 3.4% lower at 24 hpi compared to the control, corroborating the ion leakage results and macroscopic observation that the switch from biotrophy to necrotrophy was in the process of occurring (*p* > 0.05, Figure 2b). At 72 hpi we found that, compared to the control, 51% fewer cells were alive (*p* < 0.0001; Figure 2b). These data suggest that through the first 24 h, *P. medicaginis* remains biotrophic before switching to necrotrophy (Figure 2b). Based on these results, we continued with further investigation of differences in chickpea root physiology at 12, 24, and 72 hpi as they represented the biotrophic, BNS, and necrotrophic phases of infection for *P. medicaginis,* respectively. 

### 3.2. A Large Percentage of the Chickpea Genome Is Responsive to Phytophthora Infection

To investigate the transcriptomic response of chickpea roots during the three phases of *P. medicaginis* colonisation, RNA was sequenced from mock-inoculated roots, and from *P**. medicaginis* inoculated roots at 12, 24, and 72 hpi. A PCA, annotated with 95% confidence ellipses, was performed to explore the degree of transcriptional similarity between samples. PC1 explained 57% of the variation, separating the samples by inoculated versus control (Figure 3a). This indicates that the major source of variation among the samples was the presence of *P. medicaginis*. Samples from the biotrophic and BNS phases (i.e., 12 and 24 hpi, respectively) clustered together, but were distinct from mock inoculation and the 72 hpi necrotrophic phase samples. Assessment of the significantly DEGs (1 < log2FC > −1; *p* < 0.05) across all the timepoints revealed that a total of 13,568 genes, or ~45% of the chickpea genome, were significantly differentially expressed in chickpea roots following *P. medicaginis* inoculation in at least one of the timepoints (Appendix A). The expression of a number of chickpea DEGs were chosen for verification using quantitative PCR (Appendix A). The trends of gene expression patterns were corroborated using this technique.

During the biotrophic phase, 2385 chickpea genes were up-regulated, and 6331 genes were down-regulated (12 hpi; Figure 3b). The lowest number of chickpea DEGs during the timecourse were observed during the BNS phase with 2109 genes up-regulated and 4795 genes down-regulated (24 hpi; Figure 3b). The highest number of chickpea DEGs were found during the necrotrophic phase with 2708 genes up-regulated and 8984 genes down-regulated (72 hpi; Figure 3b). We then sought to investigate the expression patterns of the DEGs to identify gene expression clusters. In total, eleven clusters (i–xi) were identified. Clusters ii, iii, v-viii displayed opposing regulation between the necrotrophic phase and the biotrophic/BNS phases (Figure 4, Appendix A). Genes within cluster vi were induced during biotrophic and BNS phases while being repressed during necrotrophy with functions in signal transduction (e.g., receptor kinase) and primary metabolism (e.g., UDP-glycosyltransferase superfamily and fructose-bisphosphate aldolase; Figure 4, Appendix A). Cluster vii contained genes that were induced during necrotrophy and repressed during the earlier phases, with functions in cell growth (e.g., microtubule motor protein) and defence (e.g., callose synthase, wound responsive genes families; Figure 4, Appendix A). Clusters iv and xi contained 1397 genes that displayed significant up-regulation across the three phases and included genes with known disease function such as LysM- and chitin-domain receptor kinases, disease resistance proteins, glucan endo-1,3-beta-glucosidases and PR-4 (chitinase; Figure 4, Appendix A). Clusters i, ix, and x contained 4045 genes that displayed significant down-regulation across the three phases and included genes with known disease function such as ethylene responsive transcription factor, PR-1, glutathione-s-transferase, ABC transporters, and disease resistance proteins (Figure 4, Appendix A).

A Venn analysis found that approximately 40% of the significantly up-regulated genes were unique to a phase of infection irrespective of timepoint (red Venn diagram), a similar observation in the case of the significantly down-regulated genes (black Venn diagram; Figure 5a). GO analysis based on biological process classifications revealed that the combined unique up- and down-regulated DEGs during the biotrophic phase at 12 hpi were principally involved in protein modification (e.g., protein phosphorylation) and primary metabolic processes (e.g., tricarboxylic acid cycle and glutamine family amino acid biosynthetic process; Figure 5b, Appendix A). Further investigation revealed that these processes were also among the enriched terms for the uniquely up-regulated genes at the biotrophic phase (12 hpi), whereas the glucan biosynthetic process, the organic cyclic compound metabolic process, the sucrose biosynthetic process and the cellulose biosynthetic process were among the top significant terms enriched for the uniquely down-regulated genes (Appendix A). The top pathways with functional enrichment during the BNS phase include nucleotide biosynthesis (i.e., the purine ribonucleoside monophosphate biosynthetic process) and metabolic processes (e.g., the fructose 6-phosphate metabolic process; Figure 5b, Appendix A). In addition to the above-mentioned processes, gas transport and regulation of response to stimulus were among the most significantly enriched terms during the BNS while the pyrimidine nucleoside catabolic process, the arginine biosynthetic process and the Mo-molybdopterin cofactor biosynthetic process were among the significantly enriched terms for the uniquely down-regulated genes at this timepoint (Appendix A). During the necrotrophic phase at 72 hpi, GO analysis revealed that pathways principally involved in both primary and secondary metabolic processes and plant defence phytohormone pathways including salicylic acid signalling were significantly enriched for the combined unique genes (Figure 5b, Appendix A). The significant terms enriched for the uniquely up-regulated genes at the phase included the spermine metabolic process, response to wounding, cellular response to oxygen-containing compounds and the lipid metabolic process, whereas for the uniquely down-regulated genes, the peptide biosynthetic process, gene expression, cell redox homeostasis and intracellular transport were among the terms significantly enriched (Appendix A).

### 3.3. Chickpea Root Phytohormone Responses Do Not Conform to the Classical Leaf Model for Hemibiotrophic Pathogenesis

To gain further insight into phytohormone signalling in chickpea roots during the biotrophic to necrotrophic switch of *P. medicaginis*, an expression analysis of marker genes involved in the synthesis, signal transduction, and downstream responses of the JA, ET, and SA signalling pathways was extracted from the RNA-seq dataset. Orthologous genes involved in JA biosynthesis showed a mixture of expression across the timepoints with some *LIPOXYGENASE* (*LOX*) genes such as *Ca_19377*, *Ca_19379*, and *Ca_19380* showing increasing repression along the progression of *P. medicaginis* infection, while others such was *Ca_02923*, *Ca_18642*, and *Ca_23464* all showed universal induction during pathogenesis. Two *ALLENE OXIDE SYNTHASE* (*AOS*; *Ca_09148*, *Ca_27206*), meanwhile, exhibited higher induction during the early phases of colonisation. Within the JA signalling pathways, one defensin-like gene (*Ca _01772*), one *PR3* (*Ca_27198*) and two *PR4* (*Ca_07590*, *Ca_08525*) were highly up-regulated across the timecourse (Figure 6a). Within the ET pathway, two genes involved in the first step of biosynthesis (*ACC SYNTHASE; ACS, Ca_06539, Ca_15533*) were highly induced in the biotrophic and BNS phases while a third showed increasing expression across the timecourse (*Ca_22724*; Figure 6b). With respect to genes involved in the rate-limiting step of biosynthesis, the step controlled by *ACC OXIDASE*
*(ACO)*, four genes were most highly induced in the biotrophic and BNS phases while the remainder showed either no differential expression across the three phases, or were down-regulated. One of the key families of ethylene response factors (*ETHYLENE RESPONSE FACTOR1*; *ERF1*) showed the highest expression during biotrophy followed by attenuated induction across the BNS and necrotrophic phases (Figure 6b). Analysis of the SA pathway revealed that the genes associated with hormone biosynthesis were most heavily repressed during biotrophy (Figure 6c). Similarly, genes encoding orthologues of the SA receptor *NPR1* were not regulated during the biotrophic phases of *P. medicaginis* colonisation, but two were induced during necrotrophy (i.e., *Ca_02072*, *Ca_26412*). Inversely, however, several SA-responsive genes were induced most during the biotrophic and BNS phases: these included all *WALL-ASSOCIATED RECEPTOR KINASE70* (*WRKY70*) genes and one *PR1* (*Ca_02619*) while another *PR1* (*Ca_21489*) was down-regulated at all phases (Figure 6c). 

We metabolically quantified levels of JA, the active form of JA (JA-Isoleucine), and SA during the different infection phases while we were unable to quantify ET. Total root JA increased significantly through the biotrophic and BNS phases (56.2 ± 12.5% and 74.4 ± 18.5% above control concentrations, respectively; Figure 6d). This level then declined to a significant repression of −30 ± 7.2% at 72 hpi. The active form, JA-Ile, showed a similar trend, but the only significant repression of this hormone occurred at 72 hpi (*p* < 0.05; −41.5 ± 6.9%; Figure 6d). SA, meanwhile, was significantly repressed during the biotrophic and BNS phases (−74.1 ± 30.4% and −87.7 ± 4.8%, respectively). The levels of recovered SA were still slightly, but were non-significantly, repressed during necrotrophy.

## 4. Discussion

Hemibiotrophic pathogens, as a broad group, are able to colonise many agronomically important plant species where they threaten yield, profitability, and sustainability. While the lifestyle of hemibiotrophs may afford multiple avenues by which we could control disease development, as they show both biotrophic and necrotrophic nutritional phases, this complexity has meant greater challenge in understanding what renders plants susceptible during each phase of disease. Using a susceptible host, we characterized the chickpea–*P. medicaginis* pathosystem to better understand what transcriptomic and hormonal changes occur in a plant host of agronomic importance during colonisation by a hemibiotrophic root pathogen. Confirming our first hypothesis, RNA-sequencing analysis of root tissue revealed that downstream defence responses were largely not induced during the biotrophic phase and were induced at a higher level during the necrotrophic phase similarly to what has been observed in both roots and leaves of other plant species. However, while we observed a biphasic SA/JA accumulation during hemibiotrophic colonisation, the timing of hormones accumulation during pathogenesis was not as predicted from leaf-based hemibiotrophic model systems, thereby negating our second hypothesis.

The length of the biotrophic infection phase in hemibiotrophic pathogens appears to be genus or species specific [11,26,58]. Attard and colleagues [36] found that the biotrophic infection phase of susceptible *Arabidopsis* Col-0 ecotype roots by *Phytophthora parasitica* lasted for approximately 30 hpi. The switch to necrotrophy was observed to be at 24 and 96 hpi for *P. capsici* and *P. infestans*, respectively, during the infection of detached susceptible tomato leaves [22,25]. This suggests that, while the exact timing of the BNS can vary between species, it is generally a relatively short-lived phase during colonisation of plant tissues. Variability in the duration of the biotrophic phase has been attributed to different nutrient requirements between unrelated pathogens [5,28,59], host genetics [60,61], and differences in the number and type of effectors produced by hemibiotrophic pathogens during the biotrophic phase [8]. Based on our observations of haustoria and on cell death, the biotrophic phase of *P. medicaginis* in our experimental system continued through the first 24 hpi followed by the beginnings of the necrotrophic phase from 36 hpi based on increased cell damage, hyphal ramification, sporulation, and necrosis within the roots. Therefore, *P. medicaginis* acts similarly to other hemibiotrophic pathogens when colonising the roots of chickpea, although the factors signalling the exit from biotrophy are currently unknown. 

Plants can display unique transcriptional responses to each phase of hemibiotrophic colonisation [25,26], although these phase-specific genes generally form a minority of the total number of genes differentially regulated across all stages of pathogenesis [22]. We found that during the biotrophic phase of *P. medicaginis* infection, processes involved in energy generation such as the tricarboxylic acid cycle (TCA) and sugar biosynthesis were up-regulated. These findings are different to previous findings, whereby glycolysis, a closely linked pathway to the TCA cycle, was repressed in the *Arabidopsis*/*P. parasitica* pathosystem during biotrophy while sugar metabolism was repressed in *Arabidopsis* during *V. dahliae* biotrophic infection [32,33]. Infection of susceptible detached tomato leaves by *P. capsici* featured down-regulation of receptor-like kinases, a similar observation seen in *Arabidopsis* roots during *P. parasitica* biotrophy [25,32]. Like these studies, we also found differential expression of receptor-like kinases although this large class was both induced and repressed. In tomato, infection with *Phytophthora infestans* resulted in the induction of many uncharacterized genes, glycoside hydrolase family 17 (GH17), and thaumatin-like proteins while chalcone synthases, genes involved in reactive oxygen species production and defensins were down-regulated [22]. Similar to the tomato/*P. infestans* pathosystem, we also found that GH17 was induced, although this showed increased transcription more globally across pathogenesis, while thaumatin-like proteins and peroxidases were generally repressed. While fewer studies have investigated the transcriptional regulation during the BNS, we found enrichment for purine and carbohydrate pathways as previously found in the *P. infestans* pathosystem by [22]. Similar to [11] we found differential expression of proteases during the BNS phase. However, in our study these proteases were induced across all phases of *P. medicaginis* infection. We found evidence for differential gene expression associated with carbohydrate metabolism and molybdopterin biosynthesis during the BNS phase, as opposed to these pathways preferentially being found during necrotrophy [11,22]. Molybdopterin is a required factor in the nutrition of many pathogens [62]. In our study we found induction of calmodulin encoding genes during the BNS phase which are known to regulate Ca^2+^-dependent induction of plant immunity [63]. Similar induction of calmodulin genes were induced during the BNS phase in the soybean/*P. sojae* pathosystem but contrary to our study peroxidases were generally induced [34]. Therefore, while our work with chickpea shows a number of similarities to previously described host responses to hemibiotrophic pathogens, especially in the regulation of nutrition and defence signalling, there are distinct differences in how chickpea responds during the biotrophic/BNS phases of *P. medicaginis* disease progression.

In the necrotrophic disease phase, the host transcriptome changes significantly. During the necrotrophic phase of *L. maculans*, lignin biosynthesis were induced in canola seedlings [11], while [26] also found that leaves of susceptible wheat seedlings displayed lignification during necrotrophy. Induction of the lignin pathway during disease has previously been linked to physical reinforcement of cell walls against penetration while also preventing spread of pathogenic toxins and enzymes [64]. Increased transcription of genes associated with secondary metabolism was also observed during this phase in the canola/*L. maculans* and in the sesame/*M. phaseolina* pathosystems [11,21]. We also found that genes associated with these broad functional classes were also significantly regulated in chickpea in the latter phases of infection, suggesting that necrotrophy is generally characterized by the induction of downstream plant defences in an effort to ward off the invading pathogen. In the *Arabidopsis*/*V. dahliae* interaction and maize/*Colletotrichum graminicola* interaction, peptide biosynthesis is regulated during necrotrophy [27,65]. Similarly, we found enrichment for a repression of peptide biosynthesis during this phase. These results are logical as the plant cells are dying at this stage. Together, our results show that there are more overlaps in how a host responds at the transcriptomic level during necrotrophy than during biotrophy. The similarities at this latter stage of disease are present regardless of the plant tissue attacked, the phylogentic background of the pathogen, or the basis of plant recognition of the pathogen (i.e., qualitative vs. quantitative resistance; [38,66]).

Plants make use of phytohormones to mount a defence against hemibiotrophic pathogens [12,14]. Originally characterized in plant leaves, the classical model of plant responses to hemibiotrophic pathogens found that the biotrophic phase is regulated by SA, whereas necrotrophy is regulated by JA/ET [67,68]. Supporting this model in crops, is the whole-plant infection of susceptible canola cotyledons by *L. maculans* [11]. In our system, the colonisation of chickpea roots by *P. medicaginis* led to opposing accumulation of JA and SA hormone levels, but in a manner opposite to what has been found in leaf hemibiotrophic models: JA biosynthesis was highly induced during biotrophy while concentrations of SA were repressed during biotrophy with levels returning to pre-infection levels during necrotrophy. Other root pathosystems have also shown SA-JA responses dissimilar to leaf model systems such as in the *Arabidopsis*/*P. parasitica* and *Fagus sylvatica:Phytophthora criticola* interactions [29,36]. Transcriptionally, we found that genes associated with JA, and its partner hormone ET, were more highly induced during biotrophy while SA signalling was repressed. Similar to our study, in *A. thaliana*, JA/ET are induced and SA repressed during biotrophy and the BNS phase of *F. oxysporum* [30]. Surprisingly, we found WRKY70 genes were weakly induced during biotrophic and BNS phases. Previous reports show that WRKY70 is induced by SA, suggesting that other pathways may regulate WRKY70 induction in chickpea [69]. We found that genes with GO annotations related to SA signalling showed significant enrichment at the necrotrophic phase, cogent with an increase in SA hormone accumulation. The SA-associated gene induction during the necrotrophic phase has also been described in pathosystems including maize/*Z. tritici* [26] and rose/*Diplocarpon rosae* [70]. It would be of interest, however, to determine if similar SA/JA/ET hormonal and transcriptional profiles are observed in *P. medicaginis* resistant varieties of chickpea and if, as found by [21], the speed and intensity of hormonal regulation are more of a factor in disease resistance. 

Altogether, we have demonstrated that there are both consistencies in how chickpea roots respond on a transcriptomic level to a hemibiotrophic pathogen, but also unique aspects not well characterised in other plant models. Specifically, our finding of the unexpected phytohormone regulation in chickpea roots during pathogenesis suggests that further investigation is needed to understand the role of phytohormones in negotiating pathogen defence in roots across a broader range of plant systems and argues for the importance of studying specific plant–pathogen interactions to understand and improve diseases resistance within agronomic pathosystems. 

## Figures and Tables

**Figure 1 microorganisms-10-00343-f001:**
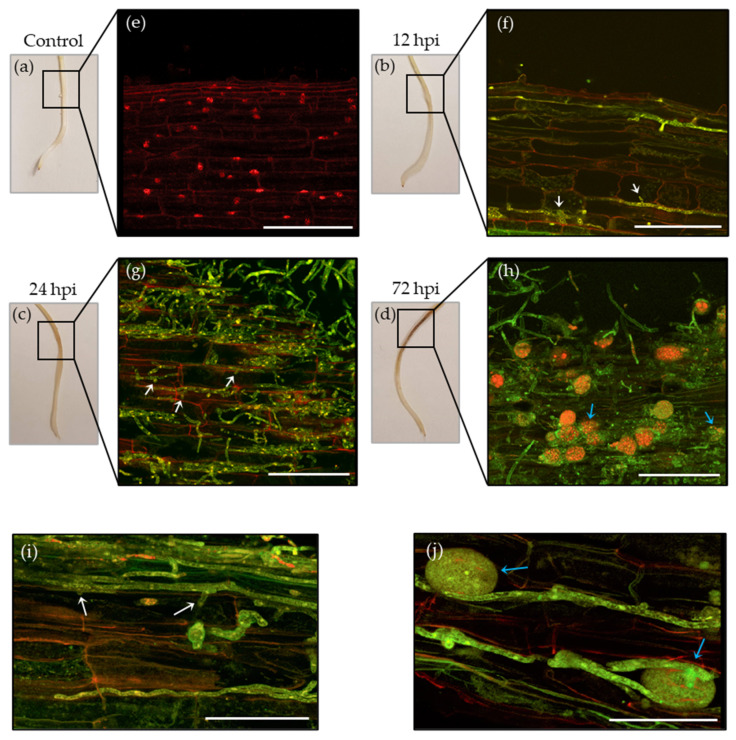
*Phytophthora medicaginis* pathogenesis is associated with a fast rate of necrosis and hemibiotrophic development. (**a**–**d**) Macroscopic symptoms of chickpea var. ‘Sonali’ either mock-inoculated with pathogen free agar blocks (control) or 12, 24 and 72 h post-inoculation (hpi) with *P. medicaginis* isolate 7831. (**e**–**h**) Confocal microscopy images of the 0.5 cm root segment that was sampled either from a mock-inoculated control or from the *P. medicaginis* inoculated site at 12, 24 and 72 hpi (sampled segment denoted by a black box). *P. medicaginis* hyphae are observed by green fluorescence while root cell walls are observed by red fluorescence. Intracellular haustoria development are indicated with white arrows while intracellular and extracellular chlamydospores are indicated by blue arrows. (**i**) Confocal microscopy images showing close up of *P. medicaginis* haustoria (white arrows) and (**j**) chlamydospores (blue arrows) photographed at 12 and 72 hpi, respectively. Scale bars: (**e**,**f**) 150 µm; (**g**,**h**) 1100 µm; (**i**) 70 µm; (**j**) 60 µm. The confocal microscopy images were taken using a TCS SP5 confocal laser scanning microscope. For imaging, a Z-stack was taken of the inoculated site using whole-roots roots for (**e**, **h**–**j**), whereas 30 µm in thickness longitudinal sections of cortical cell layer were imaged at the inoculated site for (**f**,**g**).

**Figure 2 microorganisms-10-00343-f002:**
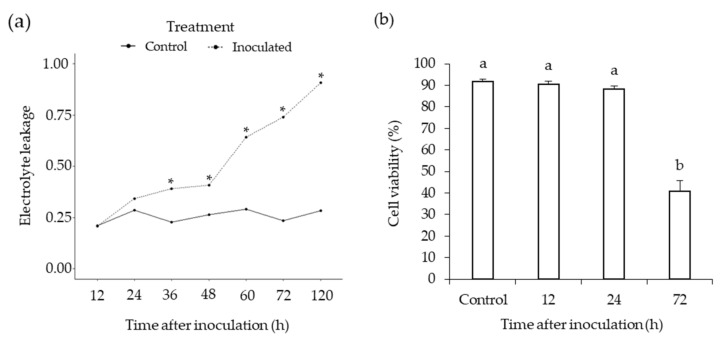
*Phytophthora medicaginis* displays a short biotrophic phase up to 24 h post-inoculation (hpi) before switching to a necrotrophic phase. (**a**) *P. medicaginis* induces significant root cell death from 36 hpi onwards. The x-axis shows the timepoints post-inoculation in hours (h) and the y-axis shows the arcsine-transformed electrolyte leakage proportions of pre-boiled relative to post-boiled root samples. * = significant differences in arcsine-transformed electrolyte leakage proportions between the inoculated and mock-inoculated control roots at a specific timepoints (*p* < 0.001, LSD = 0.08). (**b**) *P. medicaginis* causes significantly lower root cell viability at 72 hpi. The x-axis shows the control and timepoints post-inoculation in hours (h), and the y-axis shows the ratio of living to dead cells. Lower-case letters indicate significant difference between treatments (ANOVA, *p* < 0.05; *n* = 6).

**Figure 3 microorganisms-10-00343-f003:**
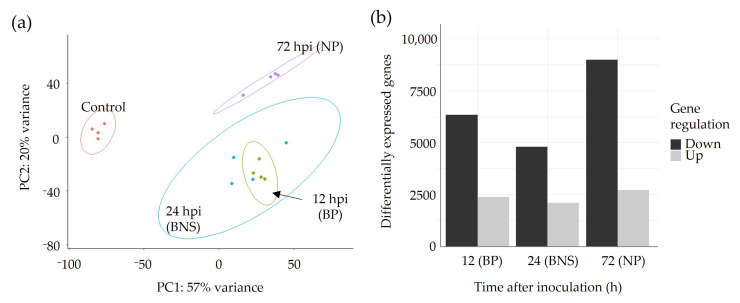
Genome-wide evaluation of RNA-sequencing data in chickpea var. ‘Sonali’ roots undergoing colonisation by *Phytophthora medicaginis*. (**a**) Principal component analysis of RNA-seq samples at 12, 24 and 72 h post-inoculation (hpi), and mock-inoculated control. (**b**) Number of significantly up-regulated and down-regulated genes in chickpea var. ‘Sonali’ roots at 12, 24 and 72 hpi (*p* < 0.05, 1 < log2FC > −1). BP: biotrophic phase, BNS: biotrophy to necrotrophy switch and NP: necrotrophic phase.

**Figure 4 microorganisms-10-00343-f004:**
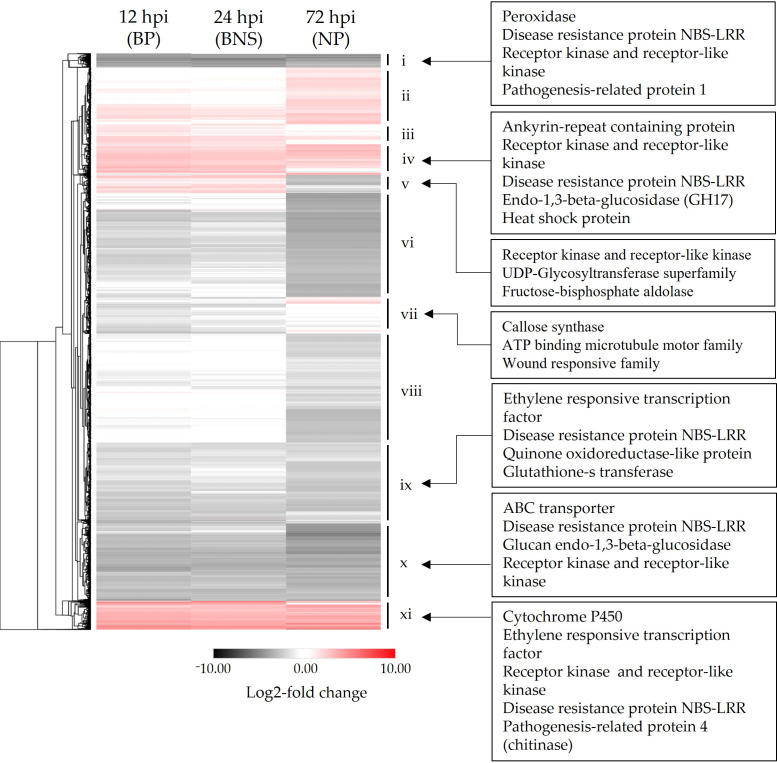
Clusters of significantly differentially regulated chickpea genes display conserved expression patterns across three phases of *Phytophthora medicaginis* colonisation. Hierarchical clustering of the log2-transformed data of 13,568 significantly regulated genes of chickpea variety ‘Sonali’ during the three phases of colonisation by *P. medicaginis* are represented (padj-value < 0.05, 1 < log2FC > −1). All data points are the ratio of transcript abundance in colonised roots as compared to control plants grown under the same conditions where red value indicates increased gene expression and black/grey indicates decreased gene expression. The heat map is annotated on the left-hand side with the hierarchical cluster groupings denoted i–xi. BP: biotrophic phase, BNS: biotrophy to necrotrophy switch, hpi: hours post-inoculation, NBS-LRR: nucleotide-binding site-leucine-rich repeat and NP: necrotrophic phase.

**Figure 5 microorganisms-10-00343-f005:**
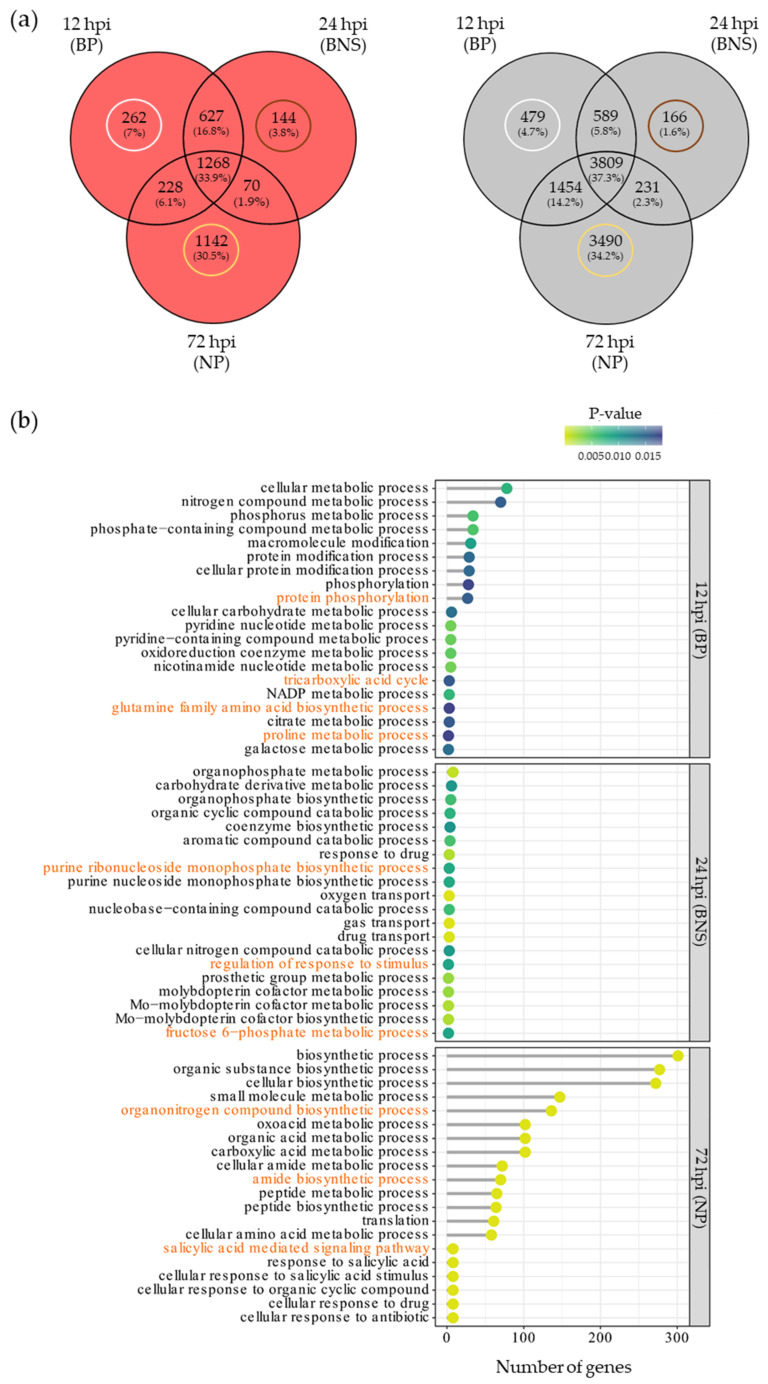
Chickpea unique responses at each of the phases reveal similar and different responses observed in other models. (**a**) Venn diagrams showing the number of common and unique significantly regulated genes in chickpea var. ‘Sonali’ varies during three timepoints of *Phytophthora medicaginis* colonisation (12, 24, and 72 h post-inoculation (hpi)), pertaining to the biotrophic phases (BP), biotrophy to necrotrophy switch (BNS) phase and necrotrophic phase (NP), respectively. The red and black Venn diagrams show the common and unique genes pertaining to the significantly up-regulated and down-regulated gene sets, respectively. The white, brown and gold circles pertain to the uniquely regulated genes that were used for gene ontology (GO) enrichment analysis at a specific phase of infection. (**b**) Enriched GO biological processes associated with differentially expressed genes in Sonali roots unique to each of the three phases of *P. medicaginis* pathogenesis. The GO terms associated with each timepoint are shown on the Y axes and the number of genes associated with each GO term are shown on the X axes. The Fisher’s exact test *p*-value statistic for each GO term are shown based on a blue–yellow gradient. GO terms discussed within the text are highlighted in orange.

**Figure 6 microorganisms-10-00343-f006:**
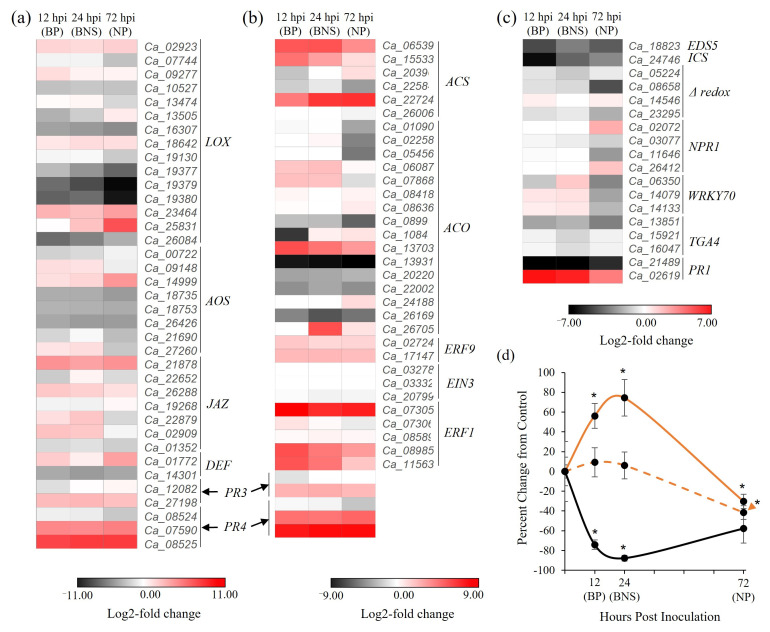
Jasmonic acid (JA), ethylene (ET), and salycilic acid (SA) signalling pathways in chickpea roots during the biotrophic to necrotrophic switch of *Phytophthora medicaginis*. (**a**) Expression profiles of the log2-transformed data of differentially regulated genes pertaining to the JA signalling pathways in chickpea var. ‘Sonali’ at 12, 24 and 72 h post-inoculation (hpi). Genes are annotated with *Arabidopsis* orthologue: *LOX*, lipoxygenase (Biosynthesis)*; AOS*, allene oxide synthase (Biosynthesis); *JAZ,* jasmonate zim domain protein (Perception); *DEF*, defensin-like gene (Signalling); *PR3*, pathogenesis-related 3 (Signalling); *PR4*, pathogenesis-related 4 (Signalling). These latter two genes are also involved in ET signalling. (**b**) Expression profiles of the log2-transformed data of differentially regulated genes pertaining to the ET pathway in chickpea var. ‘Sonali’ at 12, 24, 72 hpi. Genes are annotated with the *Arabidopsis* orthologue: *ACS,* acc synthase (Biosynthesis); *ACO,* acc oxidase (Biosynthesis); *ERF9,* ethylene response factor 9 (Signalling); *EIN3,* ethylene insensitive 3 (Signalling); *ERF1,* ethylene response factor 1 (Ssignalling). (**c**) Expression profiles of the log2-transformed data of differentially regulated genes pertaining to the SA signalling pathways in chickpea variety ‘Sonali’ at 12, 24 and 72 hpi. Down-regulated genes and up-regulated genes are presented as black (low) or red (high), respectively. Genes are annotated with *Arabidopsis* orthologue: *EDS5*, enhanced disease susceptibility 5 (Biosynthesis); *ICS*, isochorismate synthase (Biosynthesis); *NPR1*, nonexpressor of pathogenesis-related 1 (Perception); *WRKY70*, wall-associated receptor kinase 70 (Signalling); *TGA4*, tcagc-binding factor 4 (Signalling); *PR1*, pathogenesis-related 1 (Signalling). (**d**) Percent change in hormone concentrations in roots undergoing colonisation by *P. medicaginis* (compared to axenically grown control roots) for JA (orange solid line), JA-Ile (orange dashed line), and SA (black solid line). The error bars indicate standard error between 9–12 biological replicates at each timepoint; * = Significant difference from control values (*p* < 0.05; Student’s *T*-test). BP: biotrophic phase, BNS: biotrophy to necrotrophy switch, NP: necrotrophic phase.

## Data Availability

The sequencing datasets generated for this study can be found at https://www.ncbi.nlm.nih.gov/geo/query/acc.cgi?acc=GSE182741 (accessed date 31 January 2022).

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
