# Peer review of "Chickpea Roots Undergoing Colonisation by Phytophthora medicaginis Exhibit Opposing Jasmonic Acid and Salicylic Acid Accumulation and Signalling Profiles to Leaf Hemibiotrophic Models"

_microorganisms, 2022, doi:10.3390/microorganisms10020343_

Round 1

Reviewer 1 Report

This study examines chickpea-P. medicaginis pathosystem to reveal what transcriptomic and hormonal changes occur in a plant host due to colonization by a hemibiotrophic root pathogen. The authors performed transcriptome analyses of root tissue during three key phases of disease progression - biotrophic, biotrophic-to-necrotrophic switch, and necrotrophic. Their results revealed that chickpea roots display some similarities in response to hemibiotroph as previously documented in plant-pathogen foliar interactions, while metabolically quantified levels of JA, JA-Isoleucine (the active form of JA), and SA showed differently alterations to what has been observed in both roots and leaves of other plant species. The manuscript is well designed and presents an interesting result that has fundamental meaning.

I have some minor comments and observations:

Compare lines 214 and 216 – what about samples taken on 72 hpi – why are they missing from second sentence?

Introduce the abbreviations when they mentioned for a first time – for example DEG – on line 253, GO – on line 278, etc.

Reviewer 2 Report

The paper deals with root responses to colonization of Cicer arietinum  by root hemibiotroph Phytophtora medicaginis.

Using Macroscopic symptoms, electrolyte leakage and cell viability, the authors establish that Phytophtora medicaginis is hemiobiotroph, with a short biotrophic phase (up to 24 hrs on their experimental conditions) before switching to necrotrophic phase.

A chickpea transcriptomic (response RNA sequencing) was then done to compare the Biotrophc phase, the Biotrophic to Necrotrophic switch phase and the necrotrophic phase. The Necrotrophic phase correlate with a highly different. GO analysis were performed. Then the authors focus on the jasmonic acid , ethylene and Salicylic acid pathways, with the use of marker. The data are the ones from the RNA seq. JA, SA and ethylene were quantified at 0, 12 24 and 72 hr post inoculation; The results are not in favor pf the responses being conform to the classical leaf model for hemibiotrophic pathogenesis, where biotrophic phase is associated with SA response. Indeed, in the model studied, the first data indicate a decrease of SA synthesis in the first 12 hrs.

I think this is a very nice paper. I have the following remaks

1/ I am sorry that no qPCR experiments were performed.

2/ it is a pity no intermediary time points between 0 and 12 were considered.

3/ Are we sure no bias is introduced by the annotation. Do Ca_21489 and Ca_02619 correspond to the PR1 that is the functional homologous of Arabidopsis PR1…. Is there only one ICS?

Besides, it would be nice to have the same analysis with a leaf hemibiotrophic pathogen on the same plant, to have a control. Would such an analysis, mainly based on homology and annotation, be conformed to the classical model for leaf hemibiotroph? Or the non expected results is due to a biais in the analysis (annotation).

Round 2

Reviewer 2 Report

the authors took my remarks into consideration